# Structural Variant Disrupting the Expression of the Remote *FOXC1* Gene in a Patient with Syndromic Complex Microphthalmia

**DOI:** 10.3390/ijms25052669

**Published:** 2024-02-25

**Authors:** Julie Plaisancié, Bertrand Chesneau, Lucas Fares-Taie, Jean-Michel Rozet, Jacmine Pechmeja, Julien Noero, Véronique Gaston, Isabelle Bailleul-Forestier, Patrick Calvas, Nicolas Chassaing

**Affiliations:** 1Laboratoire de Référence des Anomalies Malformatives de l’Œil, Institut Fédératif de Biologie, Centre Hospitalier Universitaire de Toulouse, 31300 Toulouse, France; chesneau.b@chu-toulouse.fr (B.C.); calvas.p@outlook.fr (P.C.);; 2Centre de Référence des Affections Rares en Génétique Ophtalmologique (CARGO), Centre Hospitalier Universitaire de Toulouse, 31300 Toulouse, France; 3Molecular, Cellular and Developmental Biology Unit (MCD), Centre de Biologie Intégrative (CBI), (CNRS), Université Toulouse III Paul Sabatier (UPS), Université de Toulouse, 31062 Toulouse, France; 4Laboratoire de Génétique Ophtalmologique, Institut national de la Santé et de la Recherche Médicale (INSERM) U1163, Institut Imagine, 75015 Paris, France; lucas.fares-taie@inserm.fr (L.F.-T.);; 5Service d’Ophtalmologie, Hôpital Purpan, Centre Hospitalier Universitaire de Toulouse, 31300 Toulouse, France; 6Competence Centre of Oral Diseases, Centre Hospitalier Universitaire de Toulouse, 31300 Toulouse, France; isabelle.bailleul-forestier@univ-tlse3.fr; 7Laboratoire AURAGEN, 69003 Lyon, France

**Keywords:** structural variant, inversion, *FOXC1*, Axenfeld–Rieger, microphthalmia, eye malformation

## Abstract

Ocular malformations (OMs) arise from early defects during embryonic eye development. Despite the identification of over 100 genes linked to this heterogeneous group of disorders, the genetic cause remains unknown for half of the individuals following Whole-Exome Sequencing. Diagnosis procedures are further hampered by the difficulty of studying samples from clinically relevant tissue, which is one of the main obstacles in OMs. Whole-Genome Sequencing (WGS) to screen for non-coding regions and structural variants may unveil new diagnoses for OM individuals. In this study, we report a patient exhibiting a syndromic OM with a de novo 3.15 Mb inversion in the 6p25 region identified by WGS. This balanced structural variant was located 100 kb away from the *FOXC1* gene, previously associated with ocular defects in the literature. We hypothesized that the inversion disrupts the topologically associating domain of *FOXC1* and impairs the expression of the gene. Using a new type of samples to study transcripts, we were able to show that the patient presented monoallelic expression of *FOXC1* in conjunctival cells, consistent with the abolition of the expression of the inverted allele. This report underscores the importance of investigating structural variants, even in non-coding regions, in individuals affected by ocular malformations.

## 1. Introduction

Ocular malformations (OMs) constitute a group of clinically and genetically heterogenous disorders, representing the visible outcome of early defects during eye development. This rare group of disorders represents around 1/3000 newborns [1]. They mainly manifest as ocular growth defects (microphthalmia and anophthalmia, respectively, reduced eye size or absent eye), coloboma (closure defect of the optic fissure), aniridia and anterior segment dysgeneses (ASD). These ocular anomalies can affect one or two eyes and can also be associated with each other. When associated with a structural anomaly of the eye, microphthalmia is called complex microphthalmia. A hundred genes have been implicated in this wide and overlapping ocular phenotypic spectrum [2]. Taking all these OMs together, the genetic diagnostic yield does not exceed 50% even when using Whole-Exome Sequencing (WES) and many patients remain genetically undiagnosed [2,3,4].

Indeed, the number of OM individuals lacking a genetic diagnosis stays largely unchanged after the analysis of the main known genes by Targeted- or Whole-Exome Sequencing. The identification of new genes in OMs is actually rare, typically occurring within isolated families [5]. In other words, it seems possible that most of the main genes involved in OMs have already been discovered and that the mechanisms or lesions responsible, in particular those affecting the expression regulation of these genes, remain to be identified. Whole-Genome Sequencing (WGS) therefore appears as an interesting approach in patients with a negative WES because it allows the exploration of non-coding regions as well as the detection of balanced or unbalanced structural variants (SVs) both in coding and non-coding regions [6]. Access of non-coding regions by WGS allows detecting non-coding variants that can affect the expression of known genes. For example, Bhatia S, 2013 [7] showed the impact of a single nucleotide variant (SNV) in a 150 kb distant cis-regulatory element on the *PAX6* gene expression. Moreover, Hall et al. [8] recently demonstrated the contribution of WGS in a cohort of patients with aniridia without a genetic diagnosis, with the identification of SNVs as well as SVs affecting the *PAX6* locus. The easier accessibility of WGS in routine procedures will facilitate the exploration and the understanding of the non-coding part of genomes. This is why dissecting the impact of non-coding variants on gene expression is crucial to advancing knowledge on genome structure and function.

Here, we report a patient displaying a syndromic OM in whom we identified by WGS a 3.15 Mb inversion in the 6p25 region. This SV was located 100 kb 3′ to the *FOXC1* gene. This gene is well known for its involvement in syndromic ocular defects, notably Rieger syndrome [9]. Thus, we hypothesized that the balanced SV disrupts the topologically-associating domain (TAD) where the *FOXC1* gene is contained and therefore impacts the expression of this latter. This report underlines the need to look for SVs in coding and non-coding regions of genes involved in ocular development in order to increase the yield of genetic diagnoses in patients.

## 2. Results

### 2.1. Clinical Data

The patient has no particular familial medical history, except for a marfanoid habitus (mild skeletal features, mitral prolapse) observed in his mother, which did not meet the diagnostic criteria for Marfan syndrome [10]. The pregnancy and delivery were uneventful. At 3 months old, ophthalmological examination revealed bilateral sclerocornea with left mild microphthalmia. Ocular examination under general anesthesia ruled out lens, retinal or optic nerve anomalies. A right penetrating keratoplasty was performed, but the corneal graft progressively opacified, leading to phthisis. Consequently, a right ocular prosthesis was implanted at the age of 12.

Additionally, the patient exhibited mild neurodevelopmental delay and autistic features (mainly stereotypies), with normal neurological examination. Hearing was within normal limits. Brain MRI identified bitemporal arachnoids cysts and cysts of the lucidum septum (Appendix A). He is now a 13-year-old boy schooled half in an ordinary environment and half in a sheltered structure adapted for the visually impaired.

During the clinical examination, craniofacial features were observed, including small dysplastic ears, hypertelorism and turricephaly (Figure 1). Dental anomalies comprised areas of hypomineralization, yellowish tendency teeth and chromogenic bacterial flora, although the number of tooth germs was normal. Skeletal features encompassed a slender build, bilateral elbow deformation with decreased mobility and pes planus.

### 2.2. Genetic Data

The WGS analysis highlighted a de novo inversion of 3.15 Mb: seq[GRCh38]inv(6)(p25.1p25.3)dn;NC_:g.1713354_4870965inv (Figure 2a). Breakpoints were located in two genes: *GMDS* (chr6:1713354, within intron 9) and *CDYL* (chr6:4870965, within intron 3). Remarkably, the inversion was located 100 kb 3′ of the *FOXC1* gene (Figure 2b), already known in syndromic OM [11].

Using POSTRE, we were able to provide mechanistic putative insights of the inversion: a *GMDS* “loss-of-function” and a remote effect on the *FOXC1* gene expression. No effect was predicted regarding the second breakpoint (Appendix A).

Given the OM and the unusual dental phenotype exhibited by the patient, the disruption of *FOXC1* expression seemed to be the most clinically plausible causative mechanism. To demonstrate the impact of this inversion, we used a differentially expressed alleles approach. We were able to take a sample of a clinically relevant tissue (conjunctival cells) from him and demonstrate sufficient expression of *FOXC1* to allow transcript analysis. We knew from WGS that the proband had two heterozygous benign SNVs in the *FOXC1* coding sequence (NM_001453.3:c.[1139_1141dup;1359_1361dup];[=]) both inherited from the father, thus allowing distinction with the maternal inherited allele. The study of *FOXC1* transcripts showed the abolition of the paternal allele expression (Figure 2d) and confirms our hypothesis of the involvement of this inversion downstream of *FOXC1* in the patient’s phenotype.

## 3. Discussion

In this study, we have demonstrated, for the first time, that an inversion located far from an ocular developmental gene can disrupt its expression, leading to an OM in the patient. Indeed, *FOXC1* is involved in phenotypes associated with OMs (mainly anterior segment dysgenesis and microphthalmia) and extraocular lesions [11]. This gene encodes a transcription factor with a forkhead domain that binds DNA [9]. While the Axenfeld-Rieger anomaly (OMIM#602482) is the characteristic ASD linked to *FOXC1* mutations, the existing literature has associated various ocular defects, including microphthalmia, with pathogenic variants in this gene [9,13,14]. Moreover, some of the extra-ocular features observed in our patient such as brain abnormalities align with those recently listed as associated with *FOXC1* mutations [11]. Given the correlation between the observed phenotype and the loss of expression of a *FOXC1* allele, we concluded that the inversion was the causative genetic event behind the observed phenotype in our patient.

The overall phenotype exhibited by the individual described here closely resembles the Rieger-like phenotype observed in individuals with 6p25 deletions encompassing *FOXC1* [15,16]. In both instances, affected individuals exhibit craniofacial features characterized hypertelorism, skeletal anomalies and neuro-developmental delay. This underscores the implication of *FOXC1* as a major gene of 6p25 deletion. The implication of the two genes disrupted at the breakpoints (*GMDS* and *CDYL*) is, however, difficult to demonstrate. Despite their inclusion in Online Mendelian Inheritance in Man (OMIM, https://www.omim.org/, URL accessed on 16 January 2024), these two genes have not been previously associated with Mendelian disorders. *GMDS* encodes for GDP-mannose 4,6-dehydratase, an enzyme involved in GDP–fucose synthesis [17]. *CDYL* encodes for CDY-like protein, which seems to play a role, but which is largely unknown, in epigenetic regulation [18]. Their expression is rather ubiquitous in humans (not specific to the eye). No OM was reported in animal models (Alliance of Genome Resource V.6.0.0, https://www.alliancegenome.org/, URL accessed on 16 January 2024) and no implication in eye development was described in the literature for either of these genes. Their pLI (*GMDS* = 0.99, *CDYL* = 1) and LOEUF scores (*GMDS* = 0.09, *CDYL* = 0.05) support a high intolerance to loss-of-function variants [19]. Regarding *GMDS*, only one c.740G>C variant (p.Arg247Pro) of unknown signification is listed in the Human Gene Mutation Database (http://www.hgmd.org, URL accessed on 5 December 2023) due to its unique occurrence in a patient in a large cohort of nearly 9000 patients with neurodevelopmental disorders [20]. For *CDYL*, no variants were listed in the HGMD database.

The disruption of *FOXC1* expression is likely a consequence of the inversion interfering with a TAD in which the gene is expressed during ocular development (Figure 2b,c). The in silico analysis using POSTRE indeed indicates that the inversion leads to the loss of 11 of the 13 enhancers present on the regulatory domain of *FOXC1* and a reduction of almost 70% of the total level of H3K27ac marks reported to the number of these enhancers (which reflects the regulatory activity of these enhancers) in early neural crest cells (Appendix A). Furthermore, a recent study by D’Aurizio et al. [12] has remapped enhancer–promoter long-range interactions identified in mouse neural stem cells onto the human genome. This work revealed that one of these interactions links an enhancer located within the inverted region (wTR1_1881) to the promoter of *FOXC1* (Figure 2b).

It is crucial to highlight that transcript analysis was feasible due to the expression of *FOXC1* in a readily accessible tissue, which is not commonly the case for many OM genes. Furthermore, the presence of heterozygous *FOXC1* variants in the proband significantly facilitated the study, enabling the examination of the differential expression of the two alleles.

This study underscores the significance of employing a WGS approach for OM individuals who previously yielded negative results with WES. The identification of SVs impacting the regulation of the expression of distant ocular genes, despite their intact coding sequences, appears to represent a novel category of variants warranting careful exploration. Limited data exist in the literature regarding the impact of a WGS strategy in OMs. The few published studies indicate a modest diagnosis impact of, respectively, 24% [5] and 15% [21] in two cohorts of patients with micro-anophthalmia, though SVs were not investigated. Notably, two cases of aniridia were resolved by detecting cryptic SVs in the *PAX6* locus by using long-read WGS [22].

This finding supports our working hypothesis that a significant number of individuals with OM might present variants affecting the transcriptional control of known ocular genes, but with molecular mechanisms making it difficult to identify.

Beyond direct medical applications for patients, such sequencing strategies hold the promise of advancing our comprehension of the transcriptional networks and regulatory mechanisms operative in ocular development. This has both scientific and potentially therapeutic significance, particularly when using approaches aimed at restoring the expression of the gene whose sequence is otherwise intact.

## 4. Materials and Methods

### 4.1. Clinical Evaluation

This study was designed in compliance with the Declaration of Helsinki and informed consent was obtained from the patient and his parents. Full medical and familial history was collected. Patient underwent detailed general and ophthalmological examination. Brain MRI was performed to investigate the visual tract and intracerebral structures. Array–comparative genomic hybridization (array-CGH) and WES did not detect any variant of interest.

### 4.2. Genetic Analyles

WGS was carried out within the AURAGEN sequencing platform in France. Blood genomic DNA was used to prepare the WGS library, using the Illumina TruSeq DNA PCR-Free Library Preparation Kit, according to the manufacturer’s instructions. The libraries were sequenced on a NovaSeq 6000 (Illumina, San Diego, CA, USA), as paired-end 150 bp reads, to target a minimal average sequencing depth of 20×. After demultiplexing (bcl2fastq, Illumina), sequences were aligned to the human genome GRCh38.p13 using the BWA-MEM (0.7.17). SNV calls were made with the GATK Haplotypecaller v.4. The Large Variant calls (CNV and SV) were made by combining Manta (1.6.0) and CNVnator (0.4.1). Both parental samples were shown to be the biological parents of the patient using SNVs on WGS data. Annotation was conducted with Variant Effect Predictor (v.98.3) and further filtering was performed using an in-house pipeline.

SV breakpoints were confirmed by Sanger sequencing of genomic DNA extracted from a second blood sample. The SV effects were studied using the in silico tool named POSTRE (v1.0.0) for Prediction Of STRuctural variant Effects [23]. The *FOXC1* mRNA expression was studied by RT-PCR after RNA extraction (Qiagen, Hilden, Germany) performed on conjunctival cells (EYEPRIM, Opia Technologies, Paris, France).

## Figures and Tables

**Figure 1 ijms-25-02669-f001:**
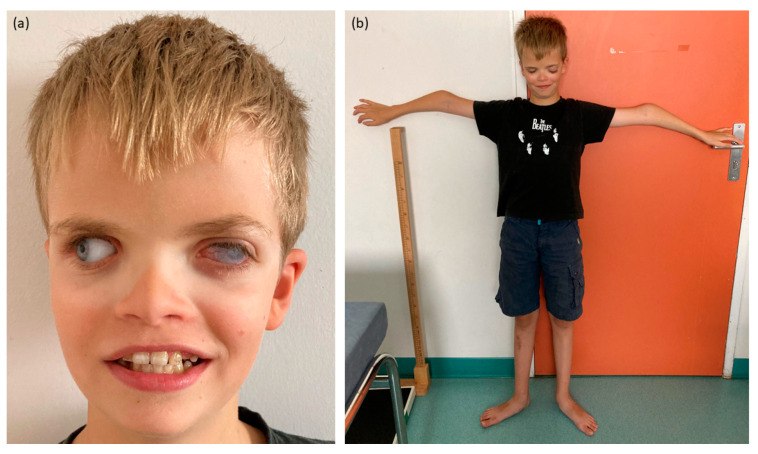
Pictures of the proband. (**a**) Craniofacial features with corneal opacity of the left eye, prosthetic right eye, hypertelorism and enamel anomalies. (**b**) Skeletal features with slender build and bilateral elbow deformation.

**Figure 2 ijms-25-02669-f002:**
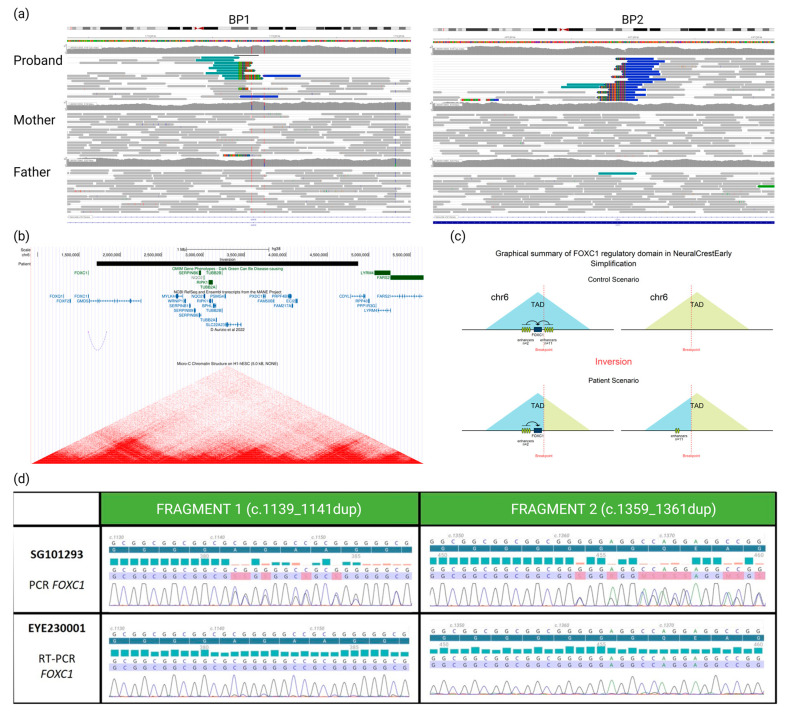
(**a**) Visualization of aligned genomic data obtained from the proband and his parents (IGV viewer) showing (in the proband only) two breakpoints (BP1 and BP2) and abnormally paired mates with higher distance and same orientation (in blue), revealing a de novo inversion of 3.15 Mb in the 6p25 region. (**b**) From Genome Browser database of the University of California, Santa Cruz (UCSC) (http://genome.ucsc.edu, URL accessed on 16 January 2024), we can see that the inversion (black rectangle) breakpoints do not disrupt any gene previously associated with a phenotype (OMIM database, green rectangles). However, BP1 disrupts *GMDS* and BP2 disrupts *CDYL* transcripts (blue line and bars). Moreover, BP1 is located 100 kb 3′ to the *FOXC1* transcript. Underneath are presented remapped interactions between enhancers and promoters, data from d’Aurizio et al., 2022 [12] and heatmaps of chromatin folding data from Micro-C XL experiments on the H1-hESC (embryonic stem cells). (**c**) Impact (predicted by POSTRE) of the inversion moving away the cis-regulatory elements from the *FOXC1* gene that they control. (**d**) We can see on the electropherograms that the patient is indeed a carrier of two heterozygous variants of the *FOXC1* gene after amplification and Sanger sequencing of two different fragments of the gene from genomic DNA (SG101293). Then, when we analyze the transcripts of this gene on RNA conjunctival sample (EYE230001), we can observe the absence of the two *FOXC1* variants and, therefore, the paternal allele carrying these two variants. Unhighlighted lines correspond to the reference sequence, the dark blue highlighted line corresponds to the reference protein sequence, blue and red bars represent PHRED quality scores of base calling (blue indicating quality > 30, red indicating quality < 20), and the blue and red highlighted line represents the sequence found in the patient, colors of the electropherogram represent the four nucleobases (cytosine (blue), guanine (black), thymine (red) and adenine (green)).

## Data Availability

All clinical and biological data relevant for the study are described in the main text and Appendix A. The novel structural variant was reported in ClinVar database (SCV004042673).

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
