# Peer review of "Structural Variant Disrupting the Expression of the Remote FOXC1 Gene in a Patient with Syndromic Complex Microphthalmia"

_ijms, 2024, doi:10.3390/ijms25052669_

Round 1

Reviewer 1 Report

Comments and Suggestions for Authors

This “Structural variant disrupting the expression of the remote FOXC1 gene in a patient with syndromic complex microphthalmia” manuscript is very interested and a great work in the field of science and discovery need some minor changes

1.) Introduction is very short authors should explain the prevalence rate of the disease at international and national level and should also explain the significance of this study for future medical research

2.) In figure 1 legend, explain both the pictures separately and label them

3.) Figure 2 (a and b) is not clearly visible improve pixel of the pictures.

4.) Are the images of brain MRI available? If yes, authors should add the pictures because they have observed that “Brain MRI identified bitemporal arachnoids cysts and cysts of the lucidum septum”.

Author Response

Reviewer 1: Comments and Suggestions for Authors

This “Structural variant disrupting the expression of the remote FOXC1 gene in a patient with syndromic complex microphthalmia” manuscript is very interested and a great work in the field of science and discovery need some minor changes

Thank you for your review and your comments.

  • Introduction is very short authors should explain the prevalence rate of the disease at international and national level and should also explain the significance of this study for future medical research

Introduction has been amended according to your comments.

  • In figure 1 legend, explain both the pictures separately and label them

As requested, legend of figure 1 has been amended.

  • Figure 2 (a and b) is not clearly visible improve pixel of the pictures.

Figure 2 has been reuploaded with better quality

  • Are the images of brain MRI available? If yes, authors should add the pictures because they have observed that “Brain MRI identified bitemporal arachnoids cysts and cysts of the lucidum septum”.

MRI images have been added as supplementary data (Figure S1).

Reviewer 2 Report

Comments and Suggestions for Authors

In the manuscript titled “Structural variant disrupting the expression of the remote FOXC1 gene in a patient with syndromic complex microphthalmia”, the author performed the whole genome sequencing on a 13-year-old patient with syndromic complex microphthalmia and reported the structural variant close to the FOXC1 gene. This study contains some valuable findings related to ocular malformations. However, current experimental design can’t support their claim that structural variant can impair the expression of FOXC1 gene. Therefore, major revision has to be done before this manuscript could be accepted for publication in the International Journal of Molecular Sciences.

Major comments

The introduction part failed to provide the detailed information for the disease (syndromic complex microphthalmia), the techniques (WGS & WES) and the genes (FOXC1 & GMDS) mentioned in the manuscript, which requires a big revision.

In the introduction part, the authors try to increase the importance of using whole genome sequencing (WGS) over whole exome sequencing (WES). Please provide some background information on these two techniques for better understandings and provide some clinical examples of the benefits using WGS over WES.

Since the structural variant is close to the FOXC1 gene, please provide more introduction for this gene, and its role in OM.

Line 83, since the structural variant make the gene GMDS loss-of-function. Please provide some information of the GMDS gene, and its relationship or potential role in OM.

The pictures in Fig 2A, 2B and 2C are not clear. Please update.

The authors performed RT-PCR to support the claim that there are two heterozygous variants of the FOXC1 gene. However, this result can’t support the claim that structural variant which is 100kb away from the gene affect its expression. As the Fig. 2B are not clear enough to get a straightforward relationship between the FOXC1 gene and the structural variant (inversion). But based on the description, it means the inversion is part of non-coding region, which can’t be analyzed from the RNA sample. If this is not true, please clarify.

Minor comments

Line 88, some typo in this sentence “we managed to take from him sample”

Please use full name of the abbreviation when it first appeared in the manuscript. For example, SNV in line 90, its full name appears in line 179.

Line 43, what is “individuals without a molecular diagnosis”, and the meaning of the whole sentence is not clear, please rewrite. Avoid using long sentences, which can confuse the readers.

Line 99, what is “UCSC website”?

Line 114, “the extra-ocular features observed in our patient”, please be specific.

Line 124, what does “OMIM” mean? Please specify.

Line 126, what is “pLI scores” and “LOEUF score”? Please specify.

Comments on the Quality of English Language

Some words in the introduction part is hard to understand, and need to be revised. Other than that, there is no problem with English quality.

Author Response

Reviewer 2: Comments and Suggestions for Authors

In the manuscript titled “Structural variant disrupting the expression of the remote FOXC1 gene in a patient with syndromic complex microphthalmia”, the author performed the whole genome sequencing on a 13-year-old patient with syndromic complex microphthalmia and reported the structural variant close to the FOXC1 gene. This study contains some valuable findings related to ocular malformations. However, current experimental design can’t support their claim that structural variant can impair the expression of FOXC1 gene. Therefore, major revision has to be done before this manuscript could be accepted for publication in the International Journal of Molecular Sciences.

Major comments

  • The introduction part failed to provide the detailed information for the disease (syndromic complex microphthalmia), the techniques (WGS & WES) and the genes (FOXC1 & GMDS) mentioned in the manuscript, which requires a big revision.

Introduction has been amended to give more details on the different aspects you mention.

  • In the introduction part, the authors try to increase the importance of using whole genome sequencing (WGS) over whole exome sequencing (WES). Please provide some background information on these two techniques for better understandings and provide some clinical examples of the benefits using WGS over WES.

Introduction has been amended to highlight the additional benefit of WGS compared to WES.

  • Since the structural variant is close to the FOXC1 gene, please provide more introduction for this gene, and its role in OM.

We have amended the content of the manuscript to give more details on the role of the FOXC1 gene.

  • Line 83, since the structural variant make the gene GMDS loss-of-function. Please provide some information of the GMDS gene, and its relationship or potential role in OM.

Discussion has been amended to add more details on GMDS and CDYL.

  • The pictures in Fig 2A, 2B and 2C are not clear. Please update.

Figure 2 has been reuploaded with higher resolution and caption has been amended for more clarity.

  • The authors performed RT-PCR to support the claim that there are two heterozygous variants of the FOXC1 gene. However, this result can’t support the claim that structural variant which is 100kb away from the gene affect its expression. As the Fig. 2B are not clear enough to get a straightforward relationship between the FOXC1 gene and the structural variant (inversion). But based on the description, it means the inversion is part of non-coding region, which can’t be analyzed from the RNA sample. If this is not true, please clarify.

Figure 2 caption has been amended for more clarity.

The structural variant (SV) identified in this study does not contain nor disrupt the coding region of the FOXC1 gene. However, we hypothesized that this SV affects the expression of FOXC1 given the presence of several pathogenic molecular criteria (the rarity of the variant, the accidental (de novo) occurrence in the patient, the in-silico disruption of a Topologically Associated Domain (TAD) distancing the FOXC1 gene from its cis-regulatory elements) in a patient with a phenotype already described in the literature in patients with a pathogenic variant in FOXC1.

The normal expression of genes can be affected by the presence of coding but also non-coding variants in distant elements: for example, Bhatia S showed in 2013 the impact of a single nucleotide variant (SNV) in a 150 kb distant cis-regulatory element on the PAX6 gene expression.

As RNA studies allow specifically studying the expression of genes, we therefore wanted to test this hypothesis by studying FOXC1 expression in a RNA sample. From WGS results, we knew that the proband had, in addition to the SV, two benign SNVs in the FOXC1 coding sequence, that allowed to distinguish the presence of the two alleles of the FOXC1 gene. By using RT-PCR in RNA extracted from conjunctival cells, we were able to detect the lack of expression of one out of the two alleles, which confirmed our hypothesis that a SV distant from the gene can affect its expression.

Minor comments

  • Line 88, some typo in this sentence “we managed to take from him sample”

Sentence has been amended.

  • Please use full name of the abbreviation when it first appeared in the manuscript. For example, SNV in line 90, its full name appears in line 179.

Abbreviations have been checked.

  • Line 43, what is “individuals without a molecular diagnosis”, and the meaning of the whole sentence is not clear, please rewrite. Avoid using long sentences, which can confuse the readers.

The “individuals without a molecular diagnosis” are individuals lacking a genetic diagnosis after molecular explorations. Sentence has been rephrased.

  • Line 99, what is “UCSC website”?

UCSC website has been amended to Genome Browser database (http://genome.ucsc.edu) of University of California, Santa Cruz (UCSC)

  • Line 114, “the extra-ocular features observed in our patient”, please be specific.

Sentence has been rephrased. Extra-ocular features notably include the brain anomalies described in the results.

  • Line 124, what does “OMIM” mean? Please specify.

OMIM has been amended to Online Mendelian Inheritance in Man (OMIM, https://www.omim.org/).

  • Line 126, what is “pLI scores” and “LOEUF score”? Please specify.

The pLi and LOEUF scores refers to “loss-of-function observed/expected upper bound fraction” described in Karczewski, Konrad J et al. “The mutational constraint spectrum quantified from variation in 141,456 humans.” Nature vol. 581,7809 (2020): 434-443. doi:10.1038/s41586-020-2308-7. This reference has been added in the manuscript.

Round 2

Reviewer 2 Report

Comments and Suggestions for Authors

The resolution of Figure 2 can be improved further. Besides that, I am satisfied with the authors' revision. No further comment.

Author Response

Thank you for your review.

We understand your concerns regarding the resolution of Figure 2 according to the PDF version of the manuscript. We have improved the resolution to 600dpi but this resolution gain was lost when including in the main manuscript. We have submitted the figure 2 separately and you can find a PDF version of the figure attached.
